# Social Connectedness and Smoking among Adolescents in Ireland: An Analysis of the Health Behaviour in Schoolchildren Study

**DOI:** 10.3390/ijerph20095667

**Published:** 2023-04-27

**Authors:** David S. Evans, Anne O’Farrell, Aishling Sheridan, Paul Kavanagh

**Affiliations:** 1National Social Inclusion Office, Health Service Executive, D20 KH63 Dublin, Ireland; 2Health Intelligence Unit, Health Service Executive, D20 DV79 Dublin, Ireland; 3Tobacco Free Ireland Programme, Health Service Executive, DO1 W596 Dublin, Ireland; 4Department of Epidemiology and Public Health, School of Population Health, Royal College of Surgeons in Ireland, D02 YN77 Dublin, Ireland

**Keywords:** social connectedness, smoking, schoolchildren, adolescents

## Abstract

Continuing progress with preventing smoking initiation is a key to the tobacco endgame. Home- and school-based social networks shape the health behaviour of children and adolescents. This study described the relationship between social connectedness and smoking behaviour in school-aged children in Ireland. The 2014 Irish Health Behaviour in School-aged Children (HBSC) surveyed self-reported smoking status and measured perceptions of social connectedness and support with validated and reliable questions across a random stratified sample of 9623 schoolchildren (aged 10–19). Overall, 8% of school-aged children reported smoking, in the last 30 days 52% reported smoking daily, and prevalence increased with age (*p* < 0.001). Compared with schoolchildren who did not smoke, perceptions of social connectedness and perceptions of support at home, from peers, and at school were significantly poorer for schoolchildren who smoked across all measures examined (*p* < 0.001). The poorest rated measures were for school connectedness and teacher support for smokers. Policies and practices that build and support positive environments for schoolchildren must continue to be prioritised if progress on preventing smoking initiation is to be sustained.

## 1. Introduction

Adolescence is a period of significant physical, psychological, and social change. Abstract thought, reasoning, problem-solving, decision-making, and value systems are developed, and adolescents strive for more independence and autonomy. During this period, an individual can become more distant from parents, with the role of peer relationships becoming more important [1]. Some find it difficult to adjust to their changing social life and association with friends, and can become vulnerable to experimentation with harmful substances such as tobacco [2,3]. They may believe that smoking initiation can enable access to social networks and some smoke to look ‘cool’ among their peer groups [3,4]. Phetphum et al. has found that one in six non-smoking youth (aged 13–15 years) are susceptible to smoking initiation [5]. Experimentation with cigarettes during adolescence is a significant problem, as adolescents could face a lifetime of dependency on nicotine, even after limited exposure. Nicotine is highly addictive and even one cigarette can start the process of addiction [6]. Most smokers start smoking before the age of 18 and continue into adulthood [7]. Early initiation of smoking increases health risks [8] and is associated with heavier smoking [9] and less likelihood of quitting [10].

Ireland, along with a number of other countries (e.g., New Zealand, UK, Scotland, USA, Canada, Australia, Sweden, Finland, Bangladesh), has shifted its focus from tobacco control to tobacco endgame, which aims to reduce smoking prevalence to minimal levels [11,12]. However, a recent increase in current smoking (smoked in last 30 days), identified between the 2015 and 2019 waves of the European School Survey Project on Alcohol and Other Drugs, highlights a need to ‘double-down’ on efforts to reduce initiation of smoking among schoolchildren in Ireland if tobacco endgame is to be achieved [13]. There is a need for policymakers and practitioners to find new and improved opportunities to better support schoolchildren in remaining smoke-free through this critical period of risk.

During adolescence, family, friends, and schools play an important role in influencing decision-making and behaviour. These form important dimensions of the concept of social connectedness. This has been defined as “a sense of belonging and subjective psychological bond that people feel in relation to groups of others [14]”. Wickramaratne et al. [15], in a scoping review, note that there is strong evidence showing that social connectedness promotes and protects physical and mental health and reduces mortality. Studies have also shown the important role that safe and supportive families, schools, and peers have on young people’s physical and mental wellbeing [16]. These have been identified by the World Health Organization (WHO) as key factors that help protect adolescents from health-compromising behaviours [17,18]. In terms of smoking, data are currently limited in an Irish context in terms of the importance of these factors. Such information would be valuable in terms of understanding the reasons why young people start smoking and developing initiatives to help prevent smoking initiation. In the context where tobacco endgame policy creates an urgent need to sustain reductions in smoking initiation, this study aimed to establish the association between perceptions of social connectedness and support with family, friends, and schools and smoking among Irish schoolchildren.

## 2. Materials and Methods

Data from the 2014 Irish Health Behaviour in School-aged Children (HBSC) survey were utilised for the study. This is a self-completion cross-sectional survey administered every four years in 44 countries. It is undertaken in collaboration with the World Health Organization and aims to provide a better understanding of young people’s health, wellbeing, and behaviours, with a focus on the social determinants of health. The HBSC uses a standardised survey of core questions for each country, with additional country-specific questions. Questions cover a range of topics including adolescent health and wellbeing, health-risk behaviours, the social environment (e.g., family, peer culture, school), and sociodemographic characteristics. The questions are agreed by all participating counties and have been subject to a number of validation studies (e.g., [19,20]). All nationally collected data are cleaned and checked in terms of quality prior to merging into an international dataset. Standardised guidelines have been developed for all participating countries [21,22,23]. This ensures that the data are valid and representative, allowing international comparisons and the monitoring of trends over time. Ethical approval for the study was obtained from the NUI Galway Research Ethics Committee.

### 2.1. Sample

Primary and post-primary Irish schools were selected using random sampling from a list provided by the Department of Education and Skills. Random sampling was stratified in terms of the population distribution across eight geographical regions (obtained from the population census [24]). School principals were contacted by post (with postal reminders and follow-up telephone calls), and those that agreed to voluntarily participate (230 schools, n = 59%) were sent questionnaires, parental consent forms, and information sheets for teachers and classrooms. Within the schools, class groups were randomly selected. This included third to sixth class pupils in primary schools and all except final-year students for post-primary schools (n = 16,013). Children that returned a signed parental consent form (by parent or legal guardian) were subsequently given a questionnaire by their teachers to complete in their classroom. Participation by children was voluntary and children could refuse to participate, only complete certain questions, or withdraw consent at any stage of the survey. Study participants were not provided with incentives to take part in the study and participating schools did not incur any financial costs. Completed questionnaires were returned anonymously by freepost. The survey was conducted anonymously and complied with the General Data Protection Regulation (GDPR) for European Union states [25]. Individual children or schools could not be identified from the returned questionnaires. For this analysis, third and fourth class students (youngest children) were subsequently excluded from the sample, as the questions on family friends, schools, and social support were either not included or contained different response choices on the abridged version of the questionnaire that this age group completed (with the exception of one question on school connectedness). 

### 2.2. Measures

Socio-demographic information on age (year and month of birth), gender, and social class (high: SC1, middle: SC2, and low: SC3) was collected. Social class was categorised using a classification system used for the Irish census of population based on the highest reported parental occupation [26].

Respondents were asked to state the number of days they had smoked in the last 30 days, the number of cigarettes per week (“every day”, “at least once a week but not every day”, “less than once a week”, and “I do not smoke”), and the frequency of smoking in the last 30 days (“not at all”, less than one cigarette per week”, “less than one cigarette per day”, 1–5 cigarettes per day”, “6–10 cigarettes per day”, “11–20 cigarettes per day”, and “more than 20 cigarettes per day”). Those that had smoked once or more often in the last 30 days were classified as smokers, which is consistent with previous research [23].

Family communication was measured by asking respondents to rate four statements about their family (“I think the important things are talked about”, “when I speak someone listens to what I have to say”, “we ask questions when we don’t understand each other”, “when there is a misunderstanding we talk it over until it is clear”). These measures have also been used by Tabak and Mazur [27] and were taken from the Family Dynamics Measure (FDMII) [28,29]. Response choices ranged from 1 (strongly agree) to 5 (strongly disagree). Responses to the four statements about family communication were summed to give a total score (Cronbach’s Alpha = 0.838). 

Family help and emotional support was measured was measured by asking respondents to rate four statements (“my family really tries to help me”, “I get the emotional help and support I need from my family”, “I can talk about problems with my family”, “my family is willing to help me make decisions”). These are four items from the 12 item Scale of Perceived Social Support [30] which have also been used by Michaelson et al. [31] and Tabak and Mazur [27]. Response choices ranged from 1 (very strongly disagree) to 7 (very strongly agree). Responses for the four statements were summed to give a total score (Cronbach’s alpha = 0.946).

Connectedness with school was measured by asking respondents to rate on a four-point scale how they felt about school at present (“I like it a lot”, “I like it a bit”, “I don’t like it very much”, “I don’t like it at all”). They were also asked to rate three statements about their school (“In our school the students take part in making the rules”, “the rules in this school are fair”, “I feel safe at this school”). Response choices ranged from 1 (strongly agree) to 5 (strongly disagree). Similar statements have been used in other studies [32,33] and were selected to encompass Brown and Evans’ [34] definition of school connectedness. How respondents felt about school was converted to a five-point scale (by multiplying individual scores by 1.25; a technique also employed by Thomson et al. [35]) and then summed with the three other statements about school (five-point scales) to give a total score (Cronbach’s alpha = 0.688).

Teacher support was measured by asking respondents to rate four statements about their teachers (“I am encouraged to express my own views in my class (es)”, “I feel that teachers accept me as I am”, “I feel that my teachers care about me as a person”, “I feel a lot of trust in my teachers”). Similar statements have been used by Walsh et al. [36], Mclellan et al. [37], Tabak and Mazur [27], and Moore et al. [38] Response choices ranged from 1 (strongly agree) to 5 (strongly disagree). These four statements were summed to give a total score (Cronbach’s alpha = 0.865). 

Supportiveness of school peers was measured by asking respondents to rate four statements (“the students in my class (es) enjoy being together”, “most of the students in my class (es) are kind and helpful”, “other students accept me as I am”). These have also been used by Erginoz et al. [39], Tabak and Mazur, [27], and Moore et al. [38] Response choices ranged from 1 (strongly agree) to 5 (strongly disagree). The statements were summed to give a total score (Cronbach’s alpha = 0.758). 

To enable comparisons between measures with total scores, each score was weighted to give a 100 point score (weight = 5.0 for family communication, connectedness with school, and teacher support; weight = 3.57 for family help and emotional support; weight = 6.67 for supportiveness of school peers). In addition, score values for teacher support, and supportiveness of school peers were reversed, so that for all scores 100 was the most favourable response.

### 2.3. Analysis

Data were disaggregated and analysed by smoking status for all children. Pearson’s chi-square and independent *t* tests were used to compare outcomes in smokers and non-smokers. Cohen’s D was calculated to determine effect size of explanatory variables. Weighted connectedness domain scores were dichotomised by classifying all cases that scored below the median value as ‘low’ and all those with a score at or above the median as ‘high’ [30]. Comparisons of weighted connectedness domain scores by smoking status (controlling for age, gender, and social class) were subsequently undertaken using binomial logistic regression. Statistical significance was determined at the 0.05 level. Exact 95% confidence intervals were calculated for proportions of binomial variables and for logistic regression adjusted odds ratios. The data were analysed in SPSS (version 25) and JMP (version 15) statistical packages.

## 3. Results 

### 3.1. Profile of Respondents

Of the selected schools, 59% agreed to take part in the study. Within participating schools, completed questionnaires were received from 85% of schoolchildren, giving a total of 13,611 respondents [23]. Following exclusion of third–fourth class (as questions were not comparable), there were 9623 school-aged children (10–18) for analysis. For those included in the study, 59% were girls and 41% boys. The mean age was 14.5 years with 58% aged less than 15 years. Despite gender disparities, the data were representative overall in terms of population distribution by region and social class [23].

### 3.2. Overall Patterns of Smoking 

Smoking prevalence (smoked at least once or more often in the last 30 days) was 8.2% (95% CI = 7.7–8.8%) among schoolchildren (aged 10–18 years). Over half of those that smoked (52%) were daily smokers, with 20% smoking weekly, and 28% smoking less often. Smoking prevalence rose with age, increasing from 8% at age 10, to 14% for 15–17 year olds, and 20% for those aged 18. This pattern is statistically significant (Pearson’s א^2^ = 674.738, df = 4, *p* < 0.001). There were gender differences in smoking prevalence (6.6% boys and 5.8% girls), but these were not statistically significant (Pearson’s א^2^ = 2.923, df = 1, *p* = 0.087). In addition, there were social class differences in smoking prevalence (6.6% boys and 5.8% girls), but these were not statistically significant (9% social class 1–2 compared to 7% for social class 5–6), but these also were not statistically significant (Pearson’s א^2^ = 3.380, df = 1, *p* = 0.066).

### 3.3. Family, School, Teacher, and Peer Scores

Table 1 shows mean overall scores (unweighted) in terms of family, school, and peers for smokers and non-smokers. All scores for smokers are poorer compared to non-smokers. These differences are statistically significant (*p* < 0.001), with the largest effect size found for family help and emotional support.

Figure 1 shows overall weighted scores in terms of family, school, teachers, and peers, stratified by smoking status. The highest score for both smokers and non-smokers is for school peer support and family communication, while the lowest scores are for school connectedness and teacher support for smokers, and school connectedness and family help and emotional support for non-smokers. Controlling for age, gender, and social class, binomial logistical regression analysis finds that smokers are 1.88 times more likely to have low family communication scores (95% CI = 1.58–2.25, *p* < 0.001); 1.8 times more likely to have low family help and emotional support scores (95% CI = 1.51–2.17, *p* < 0.001); 3.46 times more likely to have low school connectedness scores (95% CI = 2.80–4.29, *p* < 0.001); 3.17 times more likely to have low teacher support scores (95% CI = 2.26–3.28, *p* < 0.001); and 1.61 times more likely to have low school peer support scores (95% CI = 1.36–1.92, *p* < 0.001). 

## 4. Discussion 

Delivery of a policy goal of tobacco endgame in Ireland will be critically dependent on sustaining progress on preventing smoking initiation among schoolchildren. Concern regarding recent increases in smoking in this age group creates a new drive to find new opportunities to better support schoolchildren avoid the risks associated with smoking, and this study highlights social connectedness as a potential option. Through secondary analysis of the HBSC study, this study found that, compared with children who do not smoke, schoolchildren who smoke report poorer perception of social connectedness both at home and at school. This has implications in terms of the future direction of policy initiatives to prevent smoking and, indeed, potentially other risk behaviours.

Influencing schoolchildren in terms of the decision to initiate smoking is crucial if smoking prevalence is to be reduced and, as in the case of Ireland, ultimately reduced to minimal levels through so-called endgame policies. Young smokers can perceive smoking as something to help present themselves in favourable way. In a systematic review of studies of adolescent smoking, Littlecourt et al. [4] found that, in most studies, smoking was used as a means to look ‘cool’, ‘hard’, mature, or popular. Our study, by contrast, helps to counteract such beliefs, as smokers scored poorly in terms of measures of social connectedness compared to their non-smoking counterparts. This has practice implications and should guide the development of initiatives that target youth smoking and, indeed, other health-risk behaviours. It shows that initiatives that target a specific health-risk behaviour could be complemented and reinforced by those that aim to strengthen adolescent’s connectedness to family, peers, school, and teachers.

Despite progress, smoking remains a key public health challenge among schoolchildren in Ireland. The study found that 8% of school-aged children in Ireland smoked. This represents a decline since 2010 (12%) [22]. A HBSC comparison of 15 year olds in 42 countries [40] found that Ireland had the sixth lowest rate (11%). A reduction has also been experienced in Ireland among adult smokers [41]. Despite these successes, it is concerning that over half of children that currently smoke are daily smokers, with 20% smoking weekly. More recent studies highlight that there is no room for complacency, since current smoking (smoked in the past 30 days) increased in 15–16 year olds from 13.1% in 2015 to 14.4% in 2019; the increase is more pronounced in boys than girls (16.2% versus 12.8%) [13]. This change was accompanied in Ireland by an increase in the prevalence of e-cigarette use: between 2015 and 2019, e-cigarette ever use increased from 23% to 37%, while e-cigarette current use increased from 10% to 18%. While the role of e-cigarettes in causing smoking initiation in children and young people is debated, there is systematic-review-level evidence showing the e-cigarette use in this population is associated with a four-fold increase in the likelihood of smoking initiation [42]. These children are at risk of continuing to smoke daily into adulthood. This is reinforced by the fact that smoking prevalence increased with age, which has also been experienced in other countries [43]. It is also worth noting that tobacco addiction can also occur among light smokers. Oliver et al. [44] found that 18% of those who smoked less than weekly met the criteria for tobacco use disorder. It has also been found that exposure from even low levels of smoking can have a significant impact on the adolescent brain [45]. These findings suggest that there is scope to develop initiatives to raise awareness of the dangers of addiction. A study by Roditis et al. [46] found that whilst adolescents perceived cigarettes as addictive, there was a lack of recognition that addiction would mean they would find it difficult to stop smoking or would smoke for longer than they wanted.

In terms of family support, scores for smokers are significantly lower in terms of family help and emotional support and communication. Similar results have also been found in other HBSC studies [47,48], and also studies showing that supportive family environments reduced the risk of smoking, alcohol consumption, and violence [49,50,51]. By contrast, Oztekin et al. [52] found that children whose parents adopt an authoritarian parenting style are more likely to become smokers. Developing initiatives to create supportive family environments as such may help reduce the initiation of a number of health-threatening behaviours among adolescents. Population-level training programmes for parents in core parenting skills have been shown to be effective in reducing adolescent substance abuse [53,54]. Allen [55], in a systematic review of 44 studies, found that parenting interventions were effective at reducing and preventing adolescent tobacco, alcohol, and illicit substance use. The most effective interventions involved in-person sessions with both parents and youth. The value of parenting interventions was also highlighted by Ladis et al. [56], who also emphasised the importance of school-based and multiple prevention strategies There is also scope to develop web-based resources, which would help improve accessibility [57].

Along with families, the study found that connectedness to schools is important in terms of protecting children against smoking. Smokers have significantly less favourable scores in terms of connectedness to schools, and support from teachers and school peers. Indeed, of all the measures of social connectedness, the largest difference between smokers and non-smokers is for school connectedness and teacher support. These findings are similar to Aho et al. [58], who found that low school connectedness scores (perceived teacher support, disliking school, missing more than two days of school) increased the likelihood of adolescent smoking. A Welsh HBSC study [38] found that teacher relationships were associated with substance use. Gaete et al. [59] found that the risk of smoking was reduced in schools with higher school bonding. In a meta analyses of 90 studies, Rose et al. [60] found a significant protective relationship between school connectedness (school affiliation, school belonging, and attitude about school importance) and high risk substance use, violence mental health, and sexual health. It appears that if children do not have a positive experience within school, they may look for such experiences in behaviours and relationships that could place them at risk. It is unlikely that such individuals will respond to school-based smoking education programmes if they feel alienated from school. Our findings suggest that initiatives designed to create supportive school environments are likely to have a greater impact, and perhaps may help reinforce more specific measures targeting smoking and other health-risk behaviours. This approach is supported by a systematic review of school-based interventions [61], which found that broader based multicomponent school-based interventions (e.g., school policy, local community involvement in the school, parental involvement) were effective in reducing risk behaviours including smoking. A qualitative study of three schools in Australia [62] found that a whole-school approach that utilised the health-promoting schools model [63] was able to build school connectedness. This emphasised the importance of developing partnerships between staff, students, and pupils. In addition, Carrington et al. [64], in a participatory action study of an Australian school, found a whole-school approach supported young people to feel connected to their school. It is clear that broad school-based initiatives need to be implemented by school policy-makers. In addition, teachers need to be given sufficient time on the school curriculum to build social connectedness, by addressing the social and emotional well-being of children [65].

In interpreting the data, it is important to consider the limitations. The study is based on data collected in 2014. As such, our findings may not reflect the most current patterns of behaviour among schoolchildren in Ireland, although as the relationship between social connectedness and smoking behaviour delineated, it is likely to be stable. Since the study was initiated, 2018 data have been made available for analysis, but this is outside the scope of the current study. It is important that future studies monitor the association of social connectedness and smoking in Ireland using subsequent waves of survey data when they becomes available. Our analysis remains novel in the construction of a detailed measure of social connectedness that has not been undertaken to date. Published data from the 2018 wave of the study (using broad indicators as opposed to the detailed measures used in the current study) suggest that the association of social connectedness and smoking has been maintained [66]. As HBSC data is cross-sectional, it is only possible to determine associative relationships between smoking and other variables. For causal interpretation, a longitudinal study design or qualitative study examining associations would be required. However, it is worth noting that our findings are supported by a Canadian HBSC study [67] of the home and school environment that undertook additional longitudinal analysis. The data are also limited because they is based on self-reports, with questionnaires completed in a classroom environment. Smoking is prohibited in Irish schools, and it is illegal to smoke under the age of eighteen. This may have affected responses to questions on smoking, and indeed other behaviours that may be perceived by children as socially undesirable to disclose to parents and teachers (despite anonymity). However, it is worth noting that the test–retest reliability of the HBSC questions relating to smoking has been shown to be good [68]. Despite these limitations, the associations established nevertheless provide an important insight into the role that the family and school support can have on smoking, which can be utilised to facilitate future policy development. Finally, since e-cigarette use was not included in the 2014 HBSC questionnaire, we did not examine the relationship between social connectedness and e-cigarette use. A recent Irish study highlights that risk and protective factors for combustible cigarette and e-cigarette use among children and young people are different [69]. Given the emerging problem of e-cigarette use among children and young people, and its relationship with smoking initiation, exploration of the role of social connectedness with e-cigarette use would be important in future studies, to determine if interventions that support social connectedness protect children against the harm of both products. 

## 5. Conclusions

Through identifying the association between smoking and family, school, and supportive school peers, this study demonstrates the important role that a supportive family and school (including teachers and school peers) can have in preventing smoking initiation among schoolchildren. It also suggests that the smoking status of schoolchildren may be a useful way to identify children that have wider social needs. There is a need to help strengthen social connectedness, both through population-level and targeted initiatives. Preventing smoking initiation among our children is an important element to achieving Irish government targets of a tobacco-free Ireland by 2025 [12].

## Figures and Tables

**Figure 1 ijerph-20-05667-f001:**
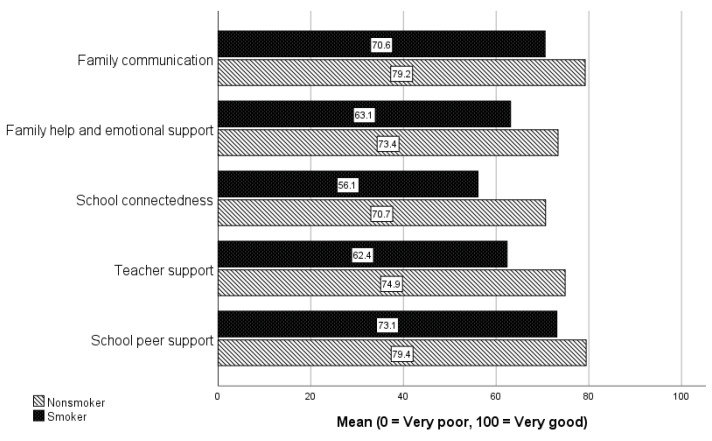
Weighted connectedness domain scores by smoking status.

**Table 1 ijerph-20-05667-t001:** Total social connectedness scores (unweighted) by smoking status.

	Smoker		Non-Smoker		Effect Size	Statistical Significance (Independent *t* Test)
	Mean	SD	Mean	SD	(Cohen’s d)	
Family communication (lower score = better communication)	9.9	4.05	8.2	3.0	3.125, CI = −0.630–−0.478	t = −11.541, df = 807.7, *p* < 0.001
Family help and emotional support (higher score = better help and emotional support)	17.7	7.9	20.6	7.8	7.845, CI = 0.289–0.443	t = 9.365 df = 8996, *p* < 0.001
Connectedness with school (lower score = better connectedness)	13.0	3.4	10.1	3.1	3.107, CI = −1.014–−0.863	t = −22.861, df = 872, *p* < 0.001
Teacher support (lower score = better support)	11.5	3.9	9.0	3.3	3.339, CI = −0.825–−0.675	t = −17.258, df = 852.5, *p* < 0.001
Supportiveness of school peers (lower score = better support)	7.0	2.5	6.1	2.1	2.127, CI = −0.519–0.371	t = −10.309, df = 871.1, *p* < 0.001

## Data Availability

Availability of data and material database are available from Department of Health Promotion, National University of Ireland Galway.

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
