# Peer review of "Social Connectedness and Smoking among Adolescents in Ireland: An Analysis of the Health Behaviour in Schoolchildren Study"

_ijerph, 2023, doi:10.3390/ijerph20095667_

Round 1

Reviewer 1 Report

The article details a study that analyzed the link between social connectedness and smoking habits in Irish schoolchildren. The study, conducted through the 2014 Irish HBSC survey, discovered that 8% of children smoke, and that smoking rates rise with age. Smokers also had lower perceived levels of social connectedness and support from family, peers, and teachers than non-smokers, with the lowest ratings being for school connectedness and teacher support. These findings highlight the need to promote positive environments in homes and schools to reduce the risk of smoking initiation and to support smoking cessation among children and adolescents.

Anyway, the article has strong limitations and needs to be improved:

Major comments:

The survey data used in the study is from 2014, and thus may not reflect the current smoking behavior of school-aged children in Ireland. In this sense, the article may lack novelty and may not be completely justified in its conclusions.

The authors should clearly describe why the survey from 2014 was used, should describe who those data are applicable currently and provide the justification of novelty. It is mentioned that this is a self-completion cross-sectional survey administered every four years in 44 countries. So it is not clear why more recent data were not analyzed. 

Minor comments:

1. Introduction must be changed, More recent investigations must me incorporated. 

2. Material and method description is vague and unclear. Please provide more detailed info. 

3. Please change the style of the discussion in order to understand how research outcomes can be used otherwise it seems that the investigation has local manner and not scientific soundness. It seems more report than scientific paper.

Please provide also the limitations of the study. 

Author Response

REVIEWER 1

We would like to thank you for taking the time to review our paper and providing constructive feedback. Based on your comments and also those of reviewer 2 we have submitted a revision of our paper for consideration. We feel that the paper is now significantly enhanced and we are grateful to you for your comments. We have listed below how we have addressed each of your comments:

 Reviewer 1 comment 1:

The survey data used in the study is from 2014, and thus may not reflect the current smoking behavior of school-aged children in Ireland. In this sense, the article may lack novelty and may not be completely justified in its conclusions.

The authors should clearly describe why the survey from 2014 was used, should describe who those data are applicable currently and provide the justification of novelty. It is mentioned that this is a self-completion cross-sectional survey administered every four years in 44 countries. So it is not clear why more recent data were not analyzed. 

Reviewer 1 comment 1 response:

The use of 2014 data has now been acknowledged as a study limitation. The novelty of our study in terms of the detailed measures of social connectedness is now emphasised:

“In interpreting the data, it is important to consider its limitations. The study is based on data collected in 2014. As such, our findings may not reflect the most current patterns of behaviour among schoolchildren in Ireland; albeit the relationship between social connectedness and smoking behaviour delineated is likely to be stable. Since the study was initiated, 2018 data has been made available for analysis, but this was outside the scope of the current study. It would be important that future studies monitor the association of social connectedness and smoking in Ireland using subsequent waves of survey data when it becomes available. Our analysis remains novel in the construction of a detailed measure of social connectedness that has not been undertaken to date. Published data from the 2018 wave of the study (using broad indicators as opposed to the detailed measures used in the current study) suggest that the association of social connectedness and smoking has been maintained [68].”

Reviewer 1 comment 2:

Introduction must be changed, More recent investigations must me incorporated. 

Reviewer 1 comment 2 response:

The introduction has been updated with more recent investigations incorporated. Over three quarters of references in the introduction were published in 2020 or later (2023 n =1, 2022 n = 6, 2021 n = 5, 2020 n= 1, 2019 n = 1, 2018 n = 2015 n = 1, 2014 n = 1, 2014 n = 1).

Reviewer 1 comment 3:

Material and method description is vague and unclear. Please provide more detailed info. 

Reviewer 1 comment 3 response:

The materials and methods section has been expanded to include more detail in terms of the HBSC survey and its administration, the sampling and ethical approval. In addition, a more detailed description of the measures used is now given. This replaces Table 1, which listed the measures used in a more summarised format.

Reviewer 1 comment 4:

Please change the style of the discussion in order to understand how research outcomes can be used otherwise it seems that the investigation has local manner and not scientific soundness. It seems more report than scientific paper.

Reviewer 1 comment 4 response:

The discussion has now been expanded and emphasis is given in terms of drawing implications in terms of how outcomes can be used. For example:

“Through secondary analysis of the HBSC study, this study found that, compared with children who do not smoke, this study identifies that schoolchildren who smoke report poorer perception of social connectedness both at home and at school. This has implications in terms of the future direction of policy initiatives to prevent smoking, and indeed potentially other risk behaviours.”

“This has practice implications and should guide the development of initiatives that target youth smoking, and indeed other health risk behaviours. It shows that initiatives that target a specific health risk behaviour could be complemented and reinforced by those that aim to strengthen adolescent’s connectedness to family, peers, school and teachers.”

“These findings suggest that there is scope to develop initiatives to raise awareness of the dangers of addiction.”

“Developing initiatives to create supportive family environments as such may help reduce the initiation of a number of health-threatening behaviours among adolescents.”

“Our findings suggest that initiatives designed at creating supportive school environments are likely to have a greater impact, and perhaps may help reinforce more specific measures targeting smoking and other health risk behaviours.”

Reviewer 1 comment 5:

Please provide also the limitations of the study. 

Reviewer 1 comment 5 response:

Study limitations are included in the discussion section. These have now been expanded  (440 words) and raises the issue of the use of cross-sectional data, the use of 2014 data, the use of self-reports, and the need to investigate the relationship between social connectedness and e-cigarette use.

Reviewer 2 Report

This paper is relevant in that it addresses a topic of interest, namely the relationship between social connectedness and smoking behaviour in school children in Ireland. The findings of the study, collected using the survey method, suggest that perceptions of social connectedness and support at home, from peers and at school were significantly worse for smokers than for non-smokers. Also, the lowest rated measures were connectedness at school and teacher support.

The title and abstract of the manuscript corresponds closely to the content of the article.

The main strengths of the manuscript are related to the social and educational interest in the chosen topic. On the other hand, the theoretical framework of the study presents an interesting review that includes primary and secondary scientific sources of interest.

In relation to the method section, it would be desirable for the authors to include more information on the procedure for recruiting participants. However, it should be noted that this study has an impressive number of participants, more than 9000 randomly selected in a stratified randomised manner within the framework of the Irish Health Behaviour in School-aged Children (HBSC) study.

Also, it should be clarified whether study participants received any kind of reward for their participation in the research. It should also be clarified whether all ethical criteria for research involving human subjects were met.

The authors should also explain the methodological procedure used. Also, in relation to the results, it is suggested that the authors provide the effect size of the analyses they have carried out.

Author Response

REVIEWER 2

We would like to thank you for taking the time to review our paper and providing constructive feedback. Based on your comments and also those of reviewer 1 we have submitted a revision of our paper for consideration. We feel that the paper is now significantly enhanced and we are grateful to you for your comments. We have listed below how we have addressed each of your comments:

Reviewer 2 comment 1:

In relation to the method section, it would be desirable for the authors to include more information on the procedure for recruiting participants. However, it should be noted that this study has an impressive number of participants, more than 9000 randomly selected in a stratified randomised manner within the framework of the Irish Health Behaviour in School-aged Children (HBSC) study.

Also, it should be clarified whether study participants received any kind of reward for their participation in the research. It should also be clarified whether all ethical criteria for research involving human subjects were met.

The authors should also explain the methodological procedure used.

Reviewer 2 comment 1 response:

The materials and methods section has been expanded to include more detail in terms of the HBSC survey, its administration and methodological procedure, the sampling, including the procedure for recruiting participants, use of rewards, and ethical approval. In addition, a more detailed description of the measures used is now given. This replaces Table 1 which listed the measures used in a more summarised format.

Reviewer 2 comment 2:

Also, in relation to the results, it is suggested that the authors provide the effect size of the analyses they have carried out.

Reviewer 2 comment 2 response:

 The effect size of the analysis is now given in Table 1

Round 2

Reviewer 1 Report

Significant changes are done. Taking into consideration that the limitations are added it can be published in current form.  

Reviewer 2 Report

I endorse the manuscript.